# Multiple-trait, random regression, and compound symmetry models for analyzing multi-environment trials in maize breeding

Igor Ferreira Coelho[1], Marco Antônio Peixoto[1], Jeniffer Santana Pinto Coelho Evangelista[1], Rodrigo Silva Alves[2], Suellen Sales[1], Marcos Deon Vilela de Resende[3], Jefferson Fernando Naves Pinto[4], Edésio Fialho dos Reis[4], Leonardo Lopes Bhering[1]*

**1** Departamento de Biologia Geral, Universidade Federal de Viçosa (UFV), Viçosa, Minas Gerais, Brazil, **2** Departamento de Estatística, INCT Café / Universidade Federal de Viçosa (UFV), Viçosa, Minas Gerais, Brazil, **3** Departamento de Estatística, Embrapa Café / Universidade Federal de Viçosa (UFV), Viçosa, Minas Gerais, Brazil, **4** Departamento de Agronomia, Universidade Federal de Jataí (UFJ), Jataí, Goiás, Brazil

* leonardo.bhering@ufv.br

**Data Availability Statement:** All relevant data are within the paper and its Supporting Information files.

## Abstract

An efficient and informative statistical method to analyze genotype-by-environment interaction (GxE) is needed in maize breeding programs. Thus, the objective of this study was to compare the effectiveness of multiple-trait models (MTM), random regression models (RRM), and compound symmetry models (CSM) in the analysis of multi-environment trials (MET) in maize breeding. For this, a data set with 84 maize hybrids evaluated across four environments for the trait grain yield (GY) was used. Variance components were estimated by restricted maximum likelihood (REML), and genetic values were predicted by best linear unbiased prediction (BLUP). The best fit MTM, RRM, and CSM were identified by the Akaike information criterion (AIC), and the significance of the genetic effects were tested using the likelihood ratio test (LRT). Genetic gains were predicted considering four selection intensities (5, 10, 15, and 20 hybrids). The selected MTM, RRM, and CSM models fit heterogeneous residuals. Moreover, for RRM the genetic effects were modeled by Legendre polynomials of order two. Genetic variability between maize hybrids were assessed for GY. In general, estimates of broad-sense heritability, selective accuracy, and predicted selection gains were slightly higher when obtained using MTM and RRM. Thus, considering the criterion of parsimony and the possibility of predicting genetic values of hybrids for untested environments, RRM is a preferential approach for analyzing MET in maize breeding.

## Introduction

Maize (*Zea mays* L.) is the most cultivated crop worldwide, with a global yield of 1.1 billion tons in the 2018/2019 crop year [1]. An important advantage of this crop is that it can be cultivated across a range of environments and seasons. However, such aspects lead to differential responses of genotypes to varied environmental conditions, which is known as genotype-by-environment interaction (GxE) [2] or phenotypic plasticity [3].

**Funding:** The authors received no specific funding for this work.

**Competing interests:** The authors have declared that no competing interests exist.

Due to the significant influence of environmental effects on the expression of quantitative traits, multi-environment trials (MET) are an important tool to assess such effects. Variations in phenotypic performance depend on the magnitude of GxE, which occurs when there is dissimilarity in a genotype's performance in different environments [4]. An environment can be defined by biotic or abiotic factors to which plants are exposed, and can include other characteristics, such as level of technology used or plant population density [5].

To assess GxE, the compound symmetry model (CSM) is the most simple and parsimonious, while the multiple-trait model (MTM) is the most complex and complete [6]. With CSM, a small number of parameters can be estimated, however the assumptions of this model are limited, such as the genetic correlation between environments being equal to 1 [7]. On the other hand, MTM tends to present problems in relation to convergence due to the large number of parameters estimated [8]. Thus, the MTM became prohibitive when the number of environments is high [6].

Random regression models (RRM) estimate the same genetic parameters as MTM but with less parameterization [9] and capture the continuous change of a trait over time or environmental gradient [8,10]. Furthermore, this approach is a powerful tool to predict reaction norms [11,12] and, consequently, predict genotypic values of genotypes for untested environments [7]. The reaction norms consider the effects of GxE and can identify the underlying causes, as they are plotted over an environmental gradient [13].

Selective accuracy is the most suitable approach to compare statistical methods in genetic improvement [14], because it is directly related to the reliability of genetic selection [15]. Moreover, selection gains and heritability are also used to compare statistical methods for analyzing GxE in maize, as maximizing selection gains is the main objective of maize breeding programs [16].

Considering the importance of GxE in maize breeding [16], effective and informative statistical methods are necessary to assess MET, with the goal of improving selective accuracy and, consequently, genetic gains with selection [8]. Recently, MTM and RRM models have been successfully used in plant breeding to analyzing the GxE [11,17]. Thus, the objective of this study was to compare MTM, RRM, and CSM in terms of analyzing MET for maize breeding.

## Material and methods

### Experimental data

The MET was carried out between January and July 2018, in Goiás State, Brazil (S1 Table). The climate of the region is classified as humid temperate (Cwa), with dry winters and hot summers [18]. The average annual temperature is around 21.5°C and average rainfall is between 1400 and 2000 mm year$^{-1}$. The agricultural practices used in the trials are based on those commonly employed for maize cultivation in the region [19].

A data set including 84 maize hybrids evaluated for the trait grain yield (GY) in four environments (E1, E2, E3, and E4) was used (S1 Table). The trials were established using a complete block design with three replications and 44 plants per plot. Plots consisted of four 4 m rows, with a spacing of 0.40 m between plants and 0.45 m between rows. To eliminate the competition effect, only the two central rows were evaluated. The GY was calculated considering the weight of the grain produced per plot, converted to kilograms per hectare (kg ha$^{-1}$).

### Statistical analyses

Three classes of statistical models, with homogeneous and heterogeneous residual variance structures, were considered: (i) CSM; (ii) MTM; and (iii) RRM fitted through Legendre polynomials. The estimation of variance components and prediction of the genotypic values for

GY were conducted via the restricted maximum likelihood/best linear unbiased prediction (REML/BLUP) procedure [20].

**(i) Compound symmetry models.** CSM was determined by the following equation:

$$y = Xr + Zg + Wge + e,$$

where: $y$ is the vector of phenotypes; $r$ is the vector of block-environment combinations (assumed to be fixed), which encompasses the effects of environment and block within environment, added to the overall mean; $g$ is the vector of genotypic effects (assumed as random); $ge$ is the vector of GxE effects (random); and $e$ is the vector of residuals (random). Uppercase letters refer to the incidence matrices for the respective effects.

In this model $g \sim N(0, I\sigma_g^2), ge \sim N(0, I\sigma_{ge}^2)$, and $e \sim N(0,R)$, where: $\sigma_g^2$ is the genotypic variance between hybrids; $\sigma_{ge}^2$ is the GxE variance; $I$ is an identity matrix with appropriate order to the respective effect; and $R$ refers to a diagonal matrix of residual variances (homogeneous or heterogeneous).

**(ii) Multiple-trait models.** MTM was determined by the following equation:

$$y = Xr + Zg + e.$$

In this model $g \sim N(0, \Sigma_g \otimes I)$ and $e \sim N(0,R)$, where: $\Sigma_g$ is the genotypic covariance matrix; $I$ is an identity matrix with the appropriate order; $\otimes$ is the Kronecker product; and $R$ refers to a diagonal matrix of residual variances (homogeneous or heterogeneous).

**(iii) Random regression models.** In order to use Legendre polynomials, the phenotypic mean of each environment ($\mu_i$) must be scaled to a range of -1 to +1. The environmental gradient values ($E_i$) were obtained with the following expression [21]:

$$E_i = -1 + 2[(\mu_i - \mu_{min})/(\mu_{max} - \mu_{min})].$$

The RRM were fitted through Legendre polynomials, considering all possible degrees of fit, as follows:

$$Y_{ijk} = \mu + S_j + R(S_{jk}) + \sum_{d=0}^{D}\alpha_{id}\Phi_{ijd} + e_{ijk},$$

where: $Y_{ijk}$ is the $i^{th}$ genotype ($i = 1, 2, \ldots, 84$) in the $j^{th}$ environment ($j = 1, 2, 3, 4$) in the $k^{th}$ block ($k = 1, 2, 3$); $\mu$ is the overall mean; $S_j$ is the fixed effect of environment $j$; $R(S_{jk})$ is the fixed effect of block $k$ within environment $j$; $d$ is the Legendre polynomial degree, ranging from $0$ to $D$ ($D = $ *number of environments—1*); $\alpha_{id}$ is the random regression coefficient for the Legendre polynomial for the genotype effects; $\Phi_{ijd}$ is the $d^{th}$ Legendre polynomial for the $j^{th}$ environment for the $i^{th}$ genotype; and $e_{ijk}$ is the residual random effect associated with $Y_{ijk}$.

In the matrix notation, the above model was described using the following equation:

$$y = Xr + Zg + e.$$

In this model $g \sim N(0, K_g \otimes I)$ and $e \sim N(0,R)$, where: $K_g$ is the covariance matrix for the coefficients of genotypic effects; $\otimes$ is the Kronecker product; $I$ is an identity matrix with the appropriate order; and $R$ refers to a diagonal matrix of residual variances (homogeneous or heterogeneous).

## Model selection

In order to compare the residual variance structures (homogeneous and heterogeneous) of CSM, MTM, and RRM the Akaike information criterion (AIC) [22] was used. The difference among the AIC values ($\Delta_{AIC}$) [23] were calculated to indicate which model provided the best

fit. Significance of the random effects of CSM, MTM, and RRM were tested using the likelihood ratio test (LRT) [24].

## Variance components and genetic parameters

For CSM, phenotypic variance ($\hat{\sigma}_p^2$), broad-sense heritability ($h^2$), coefficient of determination of GxE ($c_{ge}^2$), and genotypic correlation across environments ($r_{gloc}$), were estimated using the following expressions:

$$\hat{\sigma}_p^2 = \hat{\sigma}_g^2 + \hat{\sigma}_{ge}^2 + \hat{\sigma}_e^2,$$

$$h^2 = \frac{\hat{\sigma}_g^2}{\hat{\sigma}_p^2},$$

$$c_{ge}^2 = \frac{\hat{\sigma}_{ge}^2}{\hat{\sigma}_p^2}, \text{ and}$$

$$r_{gloc} = \frac{\hat{\sigma}_g^2}{\hat{\sigma}_g^2 + \hat{\sigma}_{ge}^2}.$$

For RRM, the estimates of genotypic variance ($\hat{\sigma}_g^2$) and predicted genotypic values ($\tilde{g}_{ij}$) at the original scale, were estimated/predicted as follows [21,25]:

$$\hat{\sigma}_g^2 = \Phi_{ijd}\hat{K}_g\Phi_{ijd}', \text{ and}$$

$$\tilde{g}_{ij} = \sum_{d=0}^{D}\hat{\alpha}_{id}\Phi_{ijd}'.$$

For MTM and RRM, phenotypic variance ($\hat{\sigma}_p^2$) and broad-sense individual heritability ($h^2$) were estimated as [6]:

$$\hat{\sigma}_p^2 = \hat{\sigma}_g^2 + \hat{\sigma}_e^2, \text{ and}$$

$$h^2 = \frac{\hat{\sigma}_g^2}{\hat{\sigma}_p^2}.$$

The selective accuracies ($r_{\hat{g}g}$) were calculated for CSM, MTM, and RRM, with the following expressions, respectively [6]:

$$r_{\hat{g}g} = \sqrt{1 - \frac{\left[1 - \left(1 - \frac{PEV}{\hat{\sigma}_g^2}\right)\right]\left[1 - \left(1 - \frac{PEV}{\hat{\sigma}_{ge}^2}\right)\right]}{1 - \left(1 - \frac{PEV}{\hat{\sigma}_g^2}\right)\left(1 - \frac{PEV}{\hat{\sigma}_{ge}^2}\right)}};$$

$$r_{\hat{g}g} = \sqrt{1 - \frac{PEV}{\hat{\sigma}_g^2}}; \text{ and}$$

$$r_{\hat{g}g} = \sqrt{1 - \frac{\Phi_{ijd}PEV\Phi_{ijd}'}{\hat{\sigma}_g^2}};$$

where *PEV* is the prediction error variance, obtained by the diagonal elements of the inverse of the coefficient matrix (information matrix) of the mixed model equations.

The experimental coefficient of variation ($CV_e$) was calculated with the following expression:

$$CV_e = \frac{\hat{\sigma}_e}{\mu_i},$$

where $\mu_i$ is the phenotypic mean of environment *i*.

### Genetic selection

The genotypes were ranked for each environment by each model. Then, selection gains (*SG*) were predicted considering the selection of 5, 10, 15, and 20 genotypes, based on the following expression [26]:

$$SG\ (\%) = \left( \frac{[(\Sigma_{i=1}^n GV_i)/n] - \mu_p}{\mu_p} \right) x\ 100,$$

where: $GV_i$ is the predicted genotypic value of genotype *i*; *n* is the number of genotypes selected; and $\mu_p$ is the overall mean.

The coincidence index was calculated to determine the similarity of the ranking of genotypes by the compared models in each environment. This index was calculated as the number of similarly ranked genotypes in the two compared models in each environment, divided by the total number of compared genotypes. The values were given in percentage for 5, 10, 15 or 20 selected genotypes, for the three models in the four environments.

### Software

Statistical analyses were performed using ASReml 4.1 Software [20] and ASReml-R package [27] of the R Software [28].

## Results

### Model selection

Based on the differences among the AIC values ($\Delta_{AIC}$), the best fit CSM was *CS.Rhe*, the best fit MTM was *MT.Rhe*, and the best fit RRM was *RR.2.Rhe*, since a difference below 2 suggests a competitive model with the best fit ($\Delta_{AIC} = 0$) [29] (Table 1). Thus, the selected models (*CS. Rhe*, *MT.Rhe*, and *RR.2.Rhe*) were used to estimate the variance components and to predict the genotypic values. Considering all models (CSM, MTM, and RRM), the best fit was *RR.2.Rhe*. In addition, according to the LRT, significant genotypic effects were detected by all models and GxE effects were detected, explicitly, by CSM (Table 1).

### Variance components and genetic parameters

Based on CSM, single estimates of genotypic and GxE variance were obtained (Table 2). On the other hand, one estimate of genotypic variance in each environment was obtained using MTM and RRM, but with no estimate of GxE variance (Table 2). For MTM and RRM, the highest estimates of genotypic variance were detected in environments E1 and E2, respectively. Environment E4 presented the lowest estimates of genotypic variance, and E3 presented the highest estimates of residual variance (Table 2).

Broad-sense heritability estimates did not follow any pattern across the evaluated environments. Considering CSM, the estimates of broad-sense heritability varied from 0.21 (E3) to

**Table 1. Model, Akaike information criterion (AIC), difference among AIC values ($\Delta_{AIC}$), and likelihood ratio test (LRT) for genotypic effects and GxE effects (in parentheses), for grain yield (GY) evaluated for 84 maize hybrids in four environments.**

| Model[a] | k[b] | AIC | $\Delta_{AIC}$ | LRT |
|---|---|---|---|---|
| CS.Rho | 3 | 14904.06 | 8.49 | 72.16** (14.11**) |
| CS.Rhe | 6 | 14895.57 | 0 | 74.06** (11.92**) |
| MT.Rho | 11 | 14888.15 | 6.44 | 211.74** |
| MT.Rhe | 14 | 14881.71 | 0 | 203.61** |
| RR.1.Rho | 2 | 14764.32 | 27.56 | 82.95** |
| RR.2.Rho | 4 | 14751.88 | 15.12 | 91.17** |
| RR.3.Rho | 7 | 14754.96 | 18.2 | 92.63** |
| RR.4.Rho | 11 | 14736.76 | 0 | 105.73** |
| RR.1.Rhe | 5 | 14753.98 | 17.22 | 81.08** |
| RR.2.Rhe | 7 | 14738.36 | 1.6 | 90.89** |
| RR.3.Rhe | 10 | 14741.68 | 4.92 | 92.23** |
| RR.4.Rhe | 14 | 14748.26 | 11.5 | 92.94** |

[a]: *CS.R_* refers to compound symmetry models, *MT.R_* refers to multiple-trait models, and *RR.O.R_* refers to random regression models, where *R_* is assumed to be homogeneous (*Rho*) or heterogenous (*Rhe*) residual variance structures, and *O* represents the Legendre polynomial order fit for the genetic effects.

[b]: Number of estimated parameters.

**: Significant at 0.01 probability of error type I by the chi-square test; the null hypothesis was that random effects did not differ from zero.

0.29 (E4), with similar results obtained for MTM and RRM (Table 2). Regarding the mean selective accuracies, a single estimate was obtained using CSM (0.88), while MTM and RRM provided one estimate for each environment (Table 2 and S2 Table). E4 was the only environment in which CSM presented a higher mean selective accuracy compared to MTM and RRM (Table 2 and S2 Table). In the other environments, the mean selective accuracies estimated by MTM or RRM were slightly higher than CSM (Table 2 and S2 Table). Based on CSM, the genotypic correlation across environments was 0.72.

## Reaction norms

The reaction norms (Fig 1), fitted through Legendre polynomials of degree two, confirmed the significance of the GxE interaction effects detected by CSM, because the reaction norms intersected, diverged, or converged (Fig 1B). Therefore, the ranking of genotypes changed across the environmental gradient (S3 Table) [30].

## Genetic selection

Based on the three selected models, the greatest selection gains were obtained in environments E1 and E2 (Table 3), while MTM provided the highest selection gains for all selection intensities (5, 10, 15, and 20 hybrids). In general, the predicted selection gains using RRM were slightly lower than those predicted by MTM and higher than those predicted by CSM. As expected, selection gains decreased as the number of selected genotypes increased (Table 3).

The coincidence indices between the selected genotypes for CSM, MTM, and RRM and each pair of environments are shown in the Fig 2.

Our results show a decrease in coincidence index values with an increase in selection intensity, and we can identify which environment had lowest coincidence values compared to other environments and across models. This information can be used to evaluate the most divergent

**Table 2. Variance components and their standard errors (in parentheses) and genetic parameters considering the compound symmetry (CSM), multiple-trait (MTM), and random regression (RRM) models for grain yield (GY) evaluated for 84 maize hybrids in four environments.**

| Component/Parameter | CSM | MTM | RRM |
|---|---|---|---|
| $\hat{\sigma}^2_{g1}$ | 324884.60 (± 67100.27) | 785802.30 (± 170462.29) | 467421.77 (± 122300.34) |
| $\hat{\sigma}^2_{g2}$ |  | 537714.80 (± 125347.84) | 533106.91 (± 148594.44) |
| $\hat{\sigma}^2_{g3}$ |  | 313339.20 (± 115232.96) | 391987.53 (± 92283.01) |
| $\hat{\sigma}^2_{g4}$ |  | 250269.40 (± 80175.67) | 226533.74 (± 29299.47) |
| $\hat{\sigma}^2_{e1}$ | 971531.00 (± 105422.84) | 852670.90 (± 94406.49) | 1054350.00 (± 104702.09) |
| $\hat{\sigma}^2_{e2}$ | 743312.90 (± 77480.13) | 752053.70 (± 82076.15) | 753868.00 (± 76071.44) |
| $\hat{\sigma}^2_{e3}$ | 1092919.30 (± 111712.22) | 1125631.50 (± 124218.84) | 1208480.00 (± 115423.11) |
| $\hat{\sigma}^2_{e4}$ | 678742.60 (± 71483.92) | 690943.60 (± 75721.01) | 702445.00 (± 77022.48) |
| $\hat{\sigma}_{ge}$ | 124696.50 (± 40296.2) | - | - |
| $h^2_1$ | 0.23 | 0.48 | 0.31 |
| $h^2_2$ | 0.27 | 0.42 | 0.41 |
| $h^2_3$ | 0.21 | 0.22 | 0.24 |
| $h^2_4$ | 0.29 | 0.27 | 0.24 |
| $c^2_{ge1}$ | 0.11 | - | - |
| $c^2_{ge2}$ | 0.14 | - | - |
| $c^2_{ge3}$ | 0.10 | - | - |
| $c^2_{ge4}$ | 0.16 | - | - |
| $\overline{r_{\hat{g}g_1}}$ | 0.88 | 0.89 | 0.91 |
| $\overline{r_{\hat{g}g_2}}$ |  | 0.91 | 0.90 |
| $\overline{r_{\hat{g}g_3}}$ |  | 0.82 | 0.91 |
| $\overline{r_{\hat{g}g_4}}$ |  | 0.83 | 0.81 |
| $CV_{e1}$ | 13.04 | 12.21 | 13.58 |
| $CV_{e2}$ | 11.01 | 11.07 | 11.09 |
| $CV_{e3}$ | 14.53 | 14.74 | 15.28 |
| $CV_{e4}$ | 14.74 | 14.87 | 14.99 |
| $r_{gloc}$ | 0.72 | - | - |
| $\mu_1$ | 7560.18 | | |
| $\mu_2$ | 7832.41 | | |
| $\mu_3$ | 7196.73 | | |
| $\mu_4$ | 5590.70 | | |

$\hat{\sigma}^2_{gi}$: Genotypic variance in environment $i$; $\hat{\sigma}^2_{ei}$: Residual variance in environment $i$; $\hat{\sigma}^2_{ge}$: GxE interaction variance; $h^2_i$: Broad-sense heritability in environment $i$; $c^2_{gei}$: Coefficient of determination of GxE interaction effect in environment $i$; $\overline{r_{\hat{g}g_i}}$: Mean selective accuracy in environment $i$; $CV_{ei}$: Experimental coefficient of variation in environment $i$; $r_{gloc}$: Genotypic correlation across environments; $\mu_i$: Phenotypic mean in environment $i$.

environment isolated and study it deeper, trying to understand its characteristics and being able to draw different breeding strategies to it.

## Discussion

### Model selection

Although both MTM and RRM enable the assessment of genetic and residual (co)variance structures [6], RRM is considered a reduced and simplified MTM, in that it provides estimates of the same genetic parameters of interest (heritability, genetic correlation), but with less parameterization and computational effort [31]. Effective modeling of genetic and residual

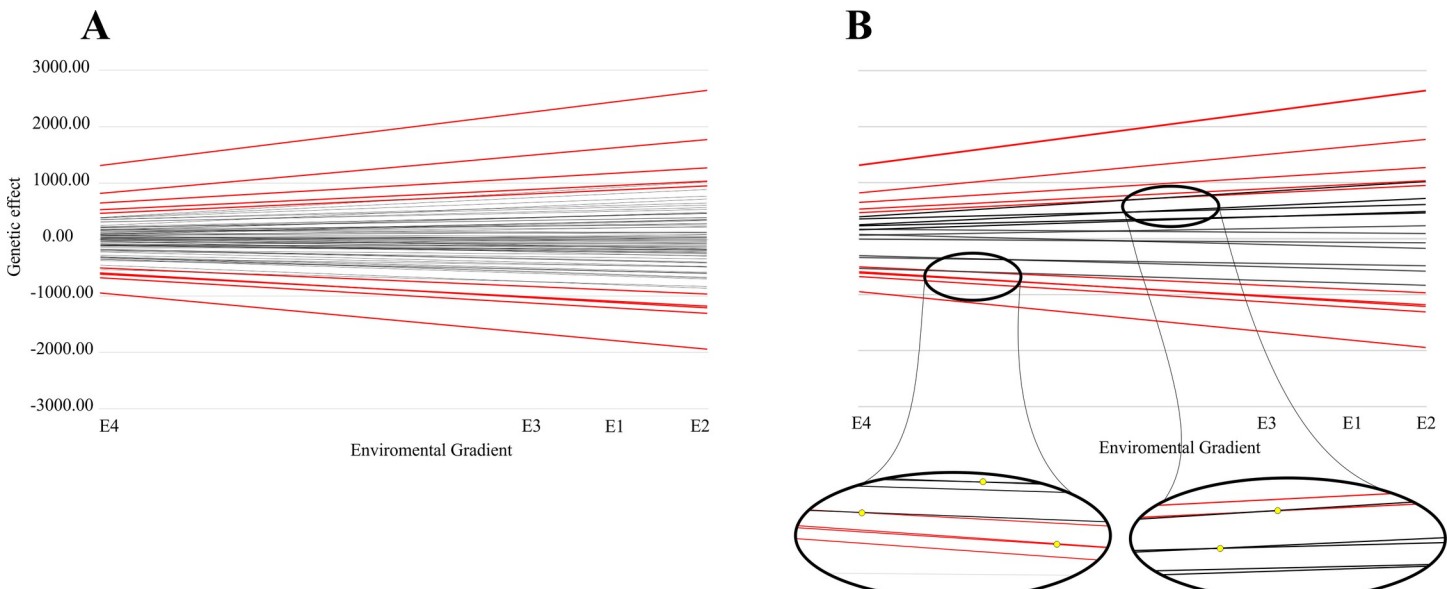

**Fig 1. Reaction norms for grain yield (GY) evaluated for maize hybrids in four environments.** "A" presents the reaction norms of the 84 maize hybrids, and "B" presents the reaction norms of around 25% of the 84 maize hybrids, and detach the reaction norms intersections using yellow dots. Both scenarios highlight the five best and worst ranked hybrids reaction norms with red color.

effects enables breeders to mitigate the adverse effects of GxE and maximize selective accuracy [32]. Resende et al. [6] highlight the importance of testing different residual variance structures, since these structures can directly affect the estimation of genetic parameters and the prediction of genetic values.

The selected models (*CS.Rhe*, *MT.Rhe*, and *RR.2.Rhe*) fit heterogeneous residuals (i.e., one residual variance for each environment). Heterogeneous residuals in MET analyses were also reported by Melo et al. [32] in analyzing progeny of the common bean, and Alves et al. [12] in

**Table 3. Predicted selection gains in percentage by the compound symmetry (CSM), multiple-trait (MTM) and random regression (RRM) models for grain yield (GY) evaluated for 84 maize hybrids in four environments.**

| Selection intensity | E1 | E2 | E3 | E4 |
|---|---|---|---|---|
| CSM | | | | |
| 5 | 21.79 | 20.07 | 13.11 | 13.49 |
| 10 | 15.65 | 14.48 | 8.95 | 10.76 |
| 15 | 13.17 | 11.02 | 7.82 | 8.92 |
| 20 | 11.43 | 9.33 | 6.45 | 6.98 |
| MTM | | | | |
| 5 | 25.54 | 20.29 | 13.91 | 15.69 |
| 10 | 19.29 | 15.35 | 10.78 | 12.26 |
| 15 | 16.05 | 12.38 | 8.92 | 10.50 |
| 20 | 13.75 | 10.56 | 7.74 | 9.12 |
| RRM | | | | |
| 5 | 25.06 | 20.29 | 13.00 | 14.74 |
| 10 | 18.86 | 15.35 | 9.78 | 11.45 |
| 15 | 14.86 | 12.29 | 8.43 | 10.27 |
| 20 | 12.95 | 10.48 | 7.16 | 8.90 |

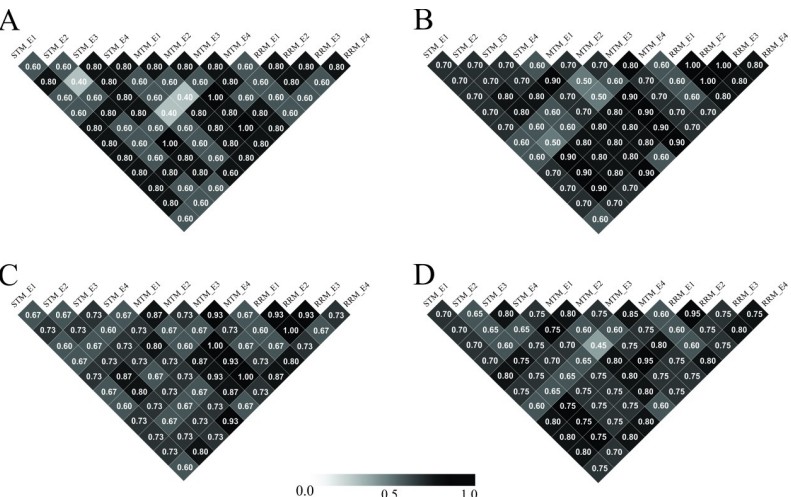

**Fig 2. Coincidence index for all environments in all models.** The compound symmetry (CSM), multiple-trait (MTM) and random regression (RRM) models are followed by each environment (E1, E2, E3, and E4). Letters A, B, C, and D correspond to each selection scenario of 5, 10, 15, and 20 hybrids, respectively.

analyzing eucalyptus clones. In this study, CSM, MTM, and RRM provided estimates for 6, 14, and 7 parameters, respectively. Thus, the parsimony (i.e., fewer parameters to be estimated) of CSM offers a significant advantage over MTM and RRM. Despite the efficacy of the Average Information REML [33], there are some issues regarding convergence for MTM (an unstructured model), particularly when the number of environments is high [6]. In the present study, RRM with the highest order (order four) was equivalent to MTM. Therefore, RRM (*RR.2.Rhe*) outperformed MTM (*MT.Rhe*) in terms of parsimony [7]. In addition, RRM enabled us to capture changes in GY continuously over the environmental gradient.

## Variance components and genetic parameters

The estimates of variance components and genetic parameters using MTM and RRM are more realistic because these models consider the heterogeneity of genetic variances among environments [6]. Based on the scale proposed by Resende and Alves [15], GY showed heritabilities of moderate magnitude ($0.15 < h^2 < 0.50$) in all environments, independently of the model used.

Selective accuracy is one of the most relevant parameters in evaluating the effectiveness of inferences of predicted genetic values [12]. This parameter is defined as the correlation between the predicted and true genotypic values and offers evidence of experimental quality, since it considers residual and genetic variances [14]. Based on Resende and Duarte [14], the mean selective accuracy magnitudes ranged from high ($0.70 < \overline{r_{\hat{g}g}} < 0.90$) to very high ($\overline{r_{\hat{g}g}} \geq 0.90$), with RRM providing very high selective accuracies in three (E1, E2, and E3) of the four evaluated environments.

The genotypic correlation across environments (0.72) estimated by CSM, underscores the need for more robust models for genetic selection [34], as CSM assumes a constant genetic variance and a correlation across environments equal to one. However, these assumptions are not prerequisites in MTM and RRM [9]; as such, the latter two models are preferred [34].

## Reaction norms

Based on CSM, GxE variance was estimated directly, and its significance was verified by LRT. For MTM and RRM, GxE variance cannot be estimated directly, since these effects are

confounded with the genetic effects ($\hat{\sigma}^2_{g_i} = \hat{\sigma}^2_{g_i} + \hat{\sigma}^2_{ge_i}$). In the random regression context, the significance of GxE effects can be determined through reaction norms; the interaction is significant when the reaction norms intersect, diverge, or converge [30]. Herein, the significance of the GxE effects is very clear because the ranking of genotypes was different across the environments, independently of the model used.

Phenotypic plasticity is essential for genotype performance in changing environments [10]. The reaction norms in the present study show that the evaluated hybrids present various forms of phenotypic plasticity. In this context, phenotypic plasticity can be considered favorable or unfavorable changes for genotype adaptedness [30].

## Genetic selection

The difference among the ranking of genotypes observed in all models is due to the GxE effects, which have an impact on genotype performance in different environments [13]. As mentioned by van Eeuwijk et al. [30], when genes are expressed differently in different environments, GxE occurs.

Regarding genetic selection, in general MTM and RRM indicate slightly higher genetic gains than the results obtained with CSM. This result can be attributed to the best statistical properties of MTM and RRM [6]. Environments E1 and E2 stood out in terms of genetic gains, when considering the same selection intensity, independently of the model used. These differences among selection gains can be explained by the estimates of broad-sense heritability, which were higher in environments E1 and E2, except for CSM in environment E4, which presented the highest estimate of broad-sense heritability. Furthermore, E1 and E2 presented the lowest coefficients of experimental variation.

The coincidence index demonstrates some characteristics of the environmental correlations. Firstly, we can see a decrease in the coincidence index with an increase in the number of selected genotypes. Secondly, the less coincident an environment $i$ is with the others, the greater the variability of an environment $i$ across the three models, and, the higher is the $CV_{ei}$ of the environment $i$, the poorer experimental quality this environment $i$ presents.

Furthermore, CSM, MTM, and RRM can be fitted through Bayesian inference or through Hierarchical Generalized BLUP (HG-BLUP) [35], both in the phenotypic and in the genomic context. Thus, relevant studies can be carried out using these approaches.

## Conclusion

The RRM presented the best fit and, consequently, provided more accurate estimates of genetic parameters and predicted genetic values. Furthermore, this model enabled to generate realistic reaction norms. Thus, the results suggest that among the three classes of statistical models, RRM is a preferential approach for analyzing MET in maize breeding.

## Supporting information

**S1 Table. Location of the four environments (E1, E2, E3, and E4) with their respective geographic coordinates and altitudes.**
(DOCX)

**S2 Table. Selective accuracy for each genotype and mean selective accuracy (below the solid line) in each environment (E1, E2, E3, and E4) based on the compound symmetry (CSM), multiple-trait (MTM), and random regression (RRM) models.**
(DOCX)

**S3 Table. Genotype ranking in each environment (E1, E2, E3, and E4), based on the compound symmetry (CSM), multiple-trait (MTM), and random regression (RRM) models.** (DOCX)

## Author Contributions

**Conceptualization:** Igor Ferreira Coelho, Marco Antônio Peixoto, Rodrigo Silva Alves, Jefferson Fernando Naves Pinto, Edésio Fialho dos Reis, Leonardo Lopes Bhering.

**Data curation:** Igor Ferreira Coelho, Marco Antônio Peixoto, Rodrigo Silva Alves.

**Formal analysis:** Igor Ferreira Coelho, Marco Antônio Peixoto, Jeniffer Santana Pinto Coelho Evangelista, Rodrigo Silva Alves.

**Investigation:** Igor Ferreira Coelho, Marco Antônio Peixoto, Suellen Sales, Jefferson Fernando Naves Pinto, Edésio Fialho dos Reis.

**Methodology:** Igor Ferreira Coelho, Marco Antônio Peixoto, Jeniffer Santana Pinto Coelho Evangelista, Rodrigo Silva Alves, Marcos Deon Vilela de Resende.

**Project administration:** Igor Ferreira Coelho, Jefferson Fernando Naves Pinto, Edésio Fialho dos Reis, Leonardo Lopes Bhering.

**Resources:** Igor Ferreira Coelho, Jefferson Fernando Naves Pinto, Edésio Fialho dos Reis, Leonardo Lopes Bhering.

**Software:** Igor Ferreira Coelho, Marco Antônio Peixoto, Jeniffer Santana Pinto Coelho Evangelista, Rodrigo Silva Alves.

**Supervision:** Rodrigo Silva Alves, Marcos Deon Vilela de Resende, Jefferson Fernando Naves Pinto, Edésio Fialho dos Reis, Leonardo Lopes Bhering.

**Validation:** Igor Ferreira Coelho, Marco Antônio Peixoto, Jeniffer Santana Pinto Coelho Evangelista.

**Visualization:** Igor Ferreira Coelho, Marco Antônio Peixoto, Rodrigo Silva Alves, Suellen Sales, Edésio Fialho dos Reis.

**Writing – original draft:** Igor Ferreira Coelho.

**Writing – review & editing:** Igor Ferreira Coelho, Marco Antônio Peixoto, Jeniffer Santana Pinto Coelho Evangelista, Rodrigo Silva Alves, Suellen Sales, Marcos Deon Vilela de Resende, Leonardo Lopes Bhering.

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
