## [Decision Letter · Decision Letter 0]

14 Jul 2020

PONE-D-20-15130

Multiple-trait, random regression, and compound symmetry models for analysis of multi-environment trials in maize breeding

PLOS ONE

Dear Dr. Bhering,

Thank you for submitting your manuscript to PLOS ONE. After careful consideration, we feel that it has merit but does not fully meet PLOS ONE’s publication criteria as it currently stands. Therefore, we invite you to submit a revised version of the manuscript that addresses the points raised during the review process.

I'm sorry it took so long for you to receive this decision, but it was really difficult to find reviewers. At the end, I was able to secure only one review, but it was quite comprehensive. I second the reviewer that this paper might have a large impact, but authors must do a much better job in accomodating the questions into the structure of the text by making it more concise and straightforward. Your main goal is relevant by itself, so avoid trying to fit in other parallel topics into the manuscript. I have provided a fully commented PDF attached.

Below I listed my major concerns:

1) The abstract needs to be fully revised. It doesn't state clearly the context of the study and provide a good summary of the results and conclusions.

2) most part of the Results is about comparing the parameters estimated in each of the three models. Since it seems important for the paper, why don't you also provide an estimate of uncertainty around each parameter? Consider fitting models under a Bayesian framework.

3) the discussion must be almost entirely re-written and restructured. It barely cites other papers in the fiels, even when there're a couple of reviews on the topic. Most of paragraphs have only one or two citations, this is not good practice in scientific writing. You have to promote a dialogue between your results and previous studies.

Most importantly, English language grammar and writing style need to be carefully revised throughout the whole manuscript. There're sentences that are really hard to understand, the reviewer also complained about it. Some parts of the text are also too much verbose, especially in the Results, try to reduce them to make it straightforward.

I hope authors are able to incorporate these suggestions to the manuscript and submit a revised version soon.

We look forward to receiving your revised manuscript.

Kind regards,

Diogo Borges Provete, PhD

Academic Editor

PLOS ONE

Journal Requirements:

"No, the funders had no role in study design, data collection and analysis, decision to

publish, or preparation of the manuscript."

Additional Editor Comments (if provided):

Reviewers' comments:

Reviewer's Responses to Questions

**Comments to the Author**

1. Is the manuscript technically sound, and do the data support the conclusions?

Reviewer #1: Partly

2. Has the statistical analysis been performed appropriately and rigorously? 

Reviewer #1: Yes

3. Have the authors made all data underlying the findings in their manuscript fully available?

Reviewer #1: Yes

4. Is the manuscript presented in an intelligible fashion and written in standard English?

Reviewer #1: Yes

5. Review Comments to the Author

Reviewer #1: Coelho et al present a detailed multi-environmental trial analysis of 84 maize hybrids. Their goal, as stated in the abstract, is to better understand the extent to which GEI interactions might be relevant to maize breeding programs. In order to do so, the authors use a model-fitting approach, comparing the model fit of three main model classes of widespread use (CSM, MTM and RRM).

Overall, the goal of this paper is clearly laid out, and the set of analyses is properly aimed at answering this key aspect of genetic variation patterns. Likewise, the amount of effort required to produce such dataset is commendable.

Having said that, this version of the manuscript also has some significant shortcomings, especially with regards to organization/clarity and in placing the work in a greater context. A significant rewrite is likely required.

In terms of organization and clarity, a substantive issue is the fact that the manuscript still tries to tackle too many issues at the same time, making it hard to follow. As made clear in the introduction, the goal of the manuscript is straightforward: compare these three model classes and infer the practical consequences of choosing one over the other (in terms of GEI). While the goal is clear, I feel that the manuscript falls short of such a goal, especially with regards to the interpretation of the results. While the manuscript reports detailed estimates of genetic variance, it does not address the ways in which all of these estimates have very different interpretations. For example, the estimate of GEI variance in the CSM model has no clear relationship to any of the parameters in the remaining models. However, it does have a potentially useful and clear interpretation. It can be not only interpreted as the portion of the total phenotypic variance that is due to GEI interactions, but can also be compared across traits or species, becoming therefore particularly useful.

The question that emerges is then, what is the practical significance of the parameters in MTM and RRM models? To what extent can they provide a intuitive understanding of the GEI interaction?

Instead, right now the manuscript goes back and forth between different topics. As such, the manuscript would greatly benefit from some streamlining. The measures of selective accuracy and selection gain are, for example, rather tangential to the main goal. Selective accuracy, in particular, is purely a theoretical expectation and follows directly from the heritability estimates. Essentially, the higher the heritability of a trait, the lower the prediction error variance. Perhaps eliminating some of these tangential aspects would help clarify the main thread. As is, the manuscript is an ensemble of variance estimates and the discussion does not help clarify their interpretation.

I would also add that the manuscript would greatly benefit from clearer figures. Most notably, Figure 1 seems to suggest that most environments have similar rankings of maize hybrids. While some hybrids have lower or higher yield across environments, their ranking seems to remain essentially the same. Instead, the remaining of the manuscript argues for the opposite of that. Given that changes in ranking have different implications than pure increases in the yield spread, clarifying this aspect of the manuscript might be particularly important. If all the hybrids that perform well in a single environment, perform (comparatively) as well in others, then GEI interactions (while still capable of generating variance) are of less importance for practical purposes, since all one would need to do is to select the ones with the highest yield at a single environment.

Finally, I would emphasize that ‘less is more’ in this case. The more clearly defined the manuscript becomes, the higher the impact it will have in its field.

Minor comment :

- The colors in most figures are hard to read and follow.

6. PLOS authors have the option to publish the peer review history of their article (what does this mean?). If published, this will include your full peer review and any attached files.

Reviewer #1: No

---

## [Author Response · Author response to Decision Letter 0]

13 Aug 2020

Dear Dr. Diogo Borges Provete, we are sending the revised version of the manuscript PONE-D-20-15130 - Multiple-trait, random regression, and compound symmetry models for analysis of multi-environment trials in maize breeding.

We would like to thank you and the reviewer for the excellent contributions to improvement of this manuscript. The changes, based on the questions and comments, were highlighted in blue (the same color of the answers in this letter).

Below we respond individually each comment and we are available for any questions.

Reviewer #1:

“…. Second the reviewer that this paper might have a large impact, but authors must do a much better job in accomodating the questions into the structure of the text by making it more concise and straightforward. Your main goal is relevant by itself, so avoid trying to fit in other parallel topics into the manuscript. I have provided a fully commented PDF attached. Below I listed my major concerns:”

1. Line 148: “Degrees of fit?”

Response: In the random regression analyses, the degrees of fit are given by the number of environments (or repeated measures) 

2. “no need to state the formula of AIC, which is widely used. Question: why didn't you use AICc instead?” …. Table 1 (line 236) “provide deltaAICc instead of raw AIC values, which are meaningless”

Response: The AIC and BIC are the standard criteria for models choice (Cavanaugh & Neath, 2019; Neath & Cavanaugh, 2012). Models with the same fixed terms are readily compared by these criteria (Verbyla, 2019). The AIC and BIC have been widely used in model selection in plant breeding data analyses (Alves et al., 2020; Melo et al., 2020). 

 We appreciate the suggestion, and we already cut off the AIC formula from the manuscript. We did not use the AICc because the n was large enough, 985. According to Brewer et al. (2016), the AICc is a correction in case of a small n, providing a stronger penalty than AIC for smaller sample. Although, Burnham and Anderson (2004) infers that AICc → AIC as n → ∞.

 3. “The abstract needs to be fully revised. It doesn't state clearly the context of the study and provide a good summary of the results and conclusions?”

 We have re-written the abstract. 

4. “Most part of the Results is about comparing the parameters estimated in each of the three models. Since it seems important for the paper, why don't you also provide an estimate of uncertainty around each parameter? Consider fitting models under a Bayesian framework.”

As commented by the other reviewer, this article is already carrying a lot of information, in fact, would be enriching to provide an uncertainty measure about the parameter, but we believed we have enough information to discuss about the parameters. The use of Bayesian inference, with no prior information, conducts to the same results (Gianola & Fernando, 1986). In addition, the implementation of these models under Bayesian framework requires more computational efforts (processing capacity and time). The idea of fit models under Bayesian framework is very good and we will consider in a future work.

5. “The discussion must be almost entirely re-written and restructured. It barely cites other papers in the fiels, even when there're a couple of reviews on the topic. Most of paragraphs have only one or two citations, this is not good practice in scientific writing. You have to promote a dialogue between your results and previous studies.”

We re-wrote and restructured the discussion as required.

6. “Most importantly, English language grammar and writing style need to be carefully revised throughout the whole manuscript. There're sentences that are really hard to understand, the reviewer also complained about it. Some parts of the text are also too much verbose, especially in the Results, try to reduce them to make it straightforward.”

We made these changes. See the Results.

Thank you very much for the review and for the excellent contributions to improvement of this manuscript.

Reviewer #2:

Coelho et al present a detailed multi-environmental trial analysis of 84 maize hybrids. Their goal, as stated in the abstract, is to better understand the extent to which GEI interactions might be relevant to maize breeding programs. In order to do so, the authors use a model-fitting approach, comparing the model fit of three main model classes of widespread use (CSM, MTM and RRM). Overall, the goal of this paper is clearly laid out, and the set of analyses is properly aimed at answering this key aspect of genetic variation patterns. Likewise, the amount of effort required to produce such dataset is commendable. Having said that, this version of the manuscript also has some significant shortcomings, especially with regards to organization/clarity and in placing the work in a greater context. A significant rewrite is likely required. In terms of organization and clarity, a substantive issue is the fact that the manuscript still tries to tackle too many issues at the same time, making it hard to follow. As made clear in the introduction, the goal of the manuscript is straightforward: compare these three model classes and infer the practical consequences of choosing one over the other (in terms of GEI). While the goal is clear, I feel that the manuscript falls short of such a goal, especially with regards to the interpretation of the results. 

1. “While the manuscript reports detailed estimates of genetic variance, it does not address the ways in which all of these estimates have very different interpretations. For example, the estimate of GEI variance in the CSM model has no clear relationship to any of the parameters in the remaining models. However, it does have a potentially useful and clear interpretation. It can be not only interpreted as the portion of the total phenotypic variance that is due to GEI interactions, but can also be compared across traits or species, becoming therefore particularly useful. The question that emerges is then, what is the practical significance of the parameters in MTM and RRM models? To what extent can they provide a intuitive understanding of the GEI interaction?”

Response: We made the changes (see Discussion, specifically lines 326-333). 

“In the random regression context, the GEI effects are detected through reaction norms (Van Eeuwijk et al., 2016). The GEI occurs when the reaction norms intersect, diverge, or converge (Van Eeuwijk et al., 2016). In the multiple-trait models, the GEI effects can not be estimated, since these effects are confounded with the genetic effects (σ ^_(g_i)^2=σ ^_(g_i)^2+σ ^_(ge_i)^2). However, if the genotype ranking differs among environments, the complex GEI occurs.”

2. “Instead, right now the manuscript goes back and forth between different topics. As such, the manuscript would greatly benefit from some streamlining. The measures of selective accuracy and selection gain are, for example, rather tangential to the main goal. Selective accuracy, in particular, is purely a theoretical expectation and follows directly

from the heritability estimates. Essentially, the higher the heritability of a trait, the lower the prediction error variance. Perhaps eliminating some of these tangential aspects would help clarify the main thread. As is, the manuscript is an ensemble of variance estimates and the discussion does not help clarify their interpretation.”

Response: The selective accuracy is the most important parameter for the genetic recommendation in plant breeding (Resende & Duarte, 2007). This parameter corresponds to the correlation between predict and true genetic values (Resende & Duarte, 2007) and it is a reliability measure.

3. “I would also add that the manuscript would greatly benefit from clearer figures. Most notably, Figure 1 seems to suggest that most environments have similar rankings of maize hybrids. While some hybrids have lower or higher yield across environments, their ranking seems to remain essentially the same. Instead, the remaining of the manuscript argues for the opposite of that. Given that changes in ranking have different implications than pure increases in the yield spread, clarifying this aspect of the manuscript might be particularly important. If all the hybrids that perform well in a single environment, perform (comparatively) as well in others, then GEI interactions (while still capable of

generating variance) are of less importance for practical purposes, since all one would need to do is to select the ones with the highest yield at a single environment.”

Response: In fact, in this study, the best genotypes in one environment are also the best in the others. However, considering all genotypes in all environments the GEI is very clear, since the reaction norms cross. We cleared the figure. Similar results were found by Alves et al. (2020) evaluating Eucalyptus clones, for the Pylodin penetration trait. Besides that, these authors verified that for traits with higher coefficients of determination of GEI variances, the genotypes raking varies more along the environmental gradient.

4. “Finally, I would emphasize that ‘less is more’ in this case. The more clearly defined the manuscript becomes, the higher the impact it will have in its field.”

 “Minor comment: - The colors in most figures are hard to read and follow.”

Response: We re-made the Results and Discuss almost entirely. (See Results and Discussion)

Thank you very much for the review and for the excellent contributions to improvement of this manuscript.

Alves, R. S., Resende, M. D. V., Azevedo, C. F., Silva, F. F., Rocha, J. R. do A. S. de C., Nunes, A. C. P., Carneiro, A. P. S., & Santos, G. A. (2020). Optimization of Eucalyptus breeding through random regression models allowing for reaction norms in response to environmental gradients. Tree Genetics & Genomes, 16(2), 38. https://doi.org/10.1007/s11295-020-01431-5

Brewer, M. J., Butler, A., & Cooksley, S. L. (2016). The relative performance of AIC, AIC C and BIC in the presence of unobserved heterogeneity. Methods in Ecology and Evolution, 7(6), 679–692. https://doi.org/10.1111/2041-210X.12541

Burnham, K. P., & Anderson, D. R. (2004). Multimodel inference: understanding AIC and BIC in model selection. Sociological Methods and Research, 33, 261–304.

Cavanaugh, J. E., & Neath, A. A. (2019). The Akaike information criterion : Background , derivation , properties , application , interpretation , and refinements. Wiley Computational Statistics, 11(January), 1–11. https://doi.org/10.1002/wics.1460

Gianola, D., & Fernando, R. L. (1986). Bayesian methods in animal breeding theory. Journal of Animal Science, 63(1), 217–244.

Melo, V. L. de, Marçal, T. de S., Rocha, J. R. A. S. de C., dos Anjos, R. S. R., Carneiro, P. C. S., & Carneiro, J. E. de S. (2020). Modeling (co)variance structures for genetic and non-genetic effects in the selection of common bean progenies. Euphytica, 216(5), 77. https://doi.org/10.1007/s10681-020-02607-9

Neath, A. A., & Cavanaugh, J. E. (2012). The Bayesian information criterion: background, derivation, and applications. Wiley Interdisciplinary Reviews: Computational Statistics, 4(2), 199–203. https://doi.org/10.1002/wics.199

Resende, M. D. V. De, & Duarte, J. B. (2007). Precisão e controle de qualidade em experimentos de avaliação de cultivares. Pesquisa Agropecuária Tropical, 37(3), 182–194. https://doi.org/10.5216/pat.v37i3.1867

Van Eeuwijk, F. A., Bustos-Korts, D. V., & Malosetti, M. (2016). What should students in plant breeding know about the statistical aspects of genotype × Environment interactions? Crop Science, 56(5), 2119–2140. https://doi.org/10.2135/cropsci2015.06.0375

Verbyla, A. P. (2019). A note on model selection using information criteria for general linear models estimated using REML. Australian and New Zealand Journal of Statistics, 61(1), 39–50. https://doi.org/10.1111/anzs.12254

---

## [Decision Letter · Decision Letter 1]

25 Sep 2020

PONE-D-20-15130R1

Multiple-trait, random regression, and compound symmetry models for analysis of multi-environment trials in maize breeding

PLOS ONE

Dear Dr. Bhering,

Thank you for submitting your manuscript to PLOS ONE. After careful consideration, we feel that it has merit but does not fully meet PLOS ONE’s publication criteria as it currently stands. Therefore, we invite you to submit a revised version of the manuscript that addresses the points raised during the review process.

Again, apologies for the delay in sending the decision. I noticed that the abstract and discussion were almost entirely re-written, as requested, and they look much clearer now. Unfortunately, I will have to agree with the reviewer. Authors did not implemented most of the changed they could in this revised version. I still believe this paper could have a big impact but the writing has to improve. Your goal is to compare methods to infer GER, then provide practical guidance to users so they can not only choose the most adequate method, but discuss their limitations and biological interpretation of model parameters. I see you replied to my concern about parameter uncertainty in the rebuttal letter, but I haven't seen any sentence about it in the actual manuscript. This is areal concern and most readers will think about it when they read your paper. So even if you're not willing to implement any Bayesian technique, at least talk about the limitations of your protocol to compare methods.

#---Specific questions:

1) You still need to provide the difference between the best model and the other models in your model selection table. This is the deltaAIC. If you include an additional colum with dAIC you don't need to indicate the best model using #, which is kind of wierd. Notice that the criteria to select models in the information theoretical framework is not the raw AIC value per se (since the AIC is dimensionless, because it's derived from the log likelihood), but the difference in AIC between competing models (see Burnham & Anderson p. 70-2). Usually a diference in AIC higher than 2 demonstrates a unequivocal support for the model with the lowest AIC.

2) The discussion still contains sentences about AIC and LRT, whcih to me should be removed. Also, some sentences of the discussion repeats parts of the methods, which doesn't make sense. Discussion is still very lengthy to the amount of results you have and must be shortened. See my comments in the pdf attached. Avoid citing tables in the discussion.

We look forward to receiving your revised manuscript.

Kind regards,

Diogo Borges Provete, PhD

Academic Editor

PLOS ONE

Additional Editor Comments (if provided):

Reviewers' comments:

Reviewer's Responses to Questions

**Comments to the Author**

1. If the authors have adequately addressed your comments raised in a previous round of review and you feel that this manuscript is now acceptable for publication, you may indicate that here to bypass the “Comments to the Author” section, enter your conflict of interest statement in the “Confidential to Editor” section, and submit your "Accept" recommendation.

Reviewer #1: (No Response)

2. Is the manuscript technically sound, and do the data support the conclusions?

Reviewer #1: Partly

3. Has the statistical analysis been performed appropriately and rigorously? 

Reviewer #1: No

4. Have the authors made all data underlying the findings in their manuscript fully available?

Reviewer #1: Yes

5. Is the manuscript presented in an intelligible fashion and written in standard English?

Reviewer #1: No

6. Review Comments to the Author

Reviewer #1: I have now reviewed the revised version of Coelho et al, in which the authors present a detailed multi-environmental trial analysis of 84 maize hybrids. As mentioned in my previous review, the goal of the paper is clear. The manuscript clearly aims to better understand the extent to which GEI interactions might be relevant to maize breeding programs. The approach is also sound. The authors use a simple model-fitting approach and directly compare the model fit of three main model classes (CSM, MTM and RRM).

Having said that, I have now read this paper multiple times and have postponed writing my review to gather my thoughts. As a reviewer, I strive to make suggestions that will allow authors to improve their manuscript. My sincere hope is that the revised version will be clearer and make a stronger argument in favor of the manuscript. In my view, the revised version of this manuscript does not succeed in addressing most of the comments raised by the editor and me in its previous iteration.

I still find the manuscript to have significant shortcomings in regard to organization/clarity and in placing the work in a greater context.

As I mentioned before, the goal of the manuscript is straightforward: compare these three model classes and infer the practical consequences of choosing one over the other (in terms of GEI). While the manuscripts presents in detail the difference between these models in terms of their estimated components, it spends little to no time exploring the practical consequences of choosing one over the other. Yes, the parameters obtained by the different models are different, but what does that mean in practical terms ? To what degree will it change the standard agricultural practice if the genotype rankings are so similar and if the metrics of selective accuracy are nearly identical between at least two of the models? Also, as mentioned by the AE, what are the uncertainties around these parameters? Uncertainty in parameters is essential for a proper interpretation of the results. Should we observe a similar type of discrepancy between the three models if we were working with another population in different environments? In other words, the manuscript needs to go beyond reporting the technical details and expand on the biological implications. As admitted in the main text of the manuscript, some of these models do not even have parameters that are capable of estimating GEI components directly (such as the MTM), despite that being the main goal of the manuscript.

Another issue that was not properly addressed was the use of figures. Other than the addition of few red lines in one of the figures (Figure 1), there was no noticeable attempt to make the figures clearer. Why not highlight the crossing interactions (if that is the main argument being made) in Figure 1? Also, the figures came without caption this time. The figure with large matrices (which has no number) is very hard to read.

In short, the revised version of Coelho et al. remains a potentially interesting contribution to the field, but falls short of making a stronger case at this point in time.

7. PLOS authors have the option to publish the peer review history of their article (what does this mean?). If published, this will include your full peer review and any attached files.

Reviewer #1: No

---

## [Author Response · Author response to Decision Letter 1]

6 Nov 2020

Response letter

Authors did not implement most of the changed they could in this revised version. I still believe this paper could have a big impact, but the writing has to improve. Your goal is to compare methods to infer GE, then provide practical guidance to users so they can not only choose the most adequate method but discuss their limitations and biological interpretation of model parameters. I see you replied to my concern about parameter uncertainty in the rebuttal letter, but I haven't seen any sentence about it in the actual manuscript. This is a real concern and most readers will think about it when they read your paper. So even if you're not willing to implement any Bayesian technique, at least talk about the limitations of your protocol to compare methods.

We agree, and we made several changes/corrections in the manuscript. The aim of this study was to compare several statistical models in the Fisherian context (REML/BLUP procedure) for MET analysis in maize breeding. Currently, REML/BLUP is the standard procedure for estimation of variance components and optimal selection in plant breeding (Resende 2016). We included one paragraph highlighting the possibilities to fit the same models through Bayesian inference or through Hierarchical Generalized BLUP (HG-BLUP). Please see the text highlight in blue in the manuscript.

Thank you very much for the review and for the excellent contributions to improvement of this manuscript.

Reviewer #1:

1. “You still need to provide the difference between the best model and the other models in your model selection table. This is the deltaAIC. If you include an additional column with dAIC you don't need to indicate the best model using #, which is kind of weird. Notice that the criteria to select models in the information theoretical framework is not the raw AIC value per se (since the AIC is dimensionless, because it's derived from the log likelihood), but the difference in AIC between competing models (see Burnham & Anderson p. 70-2). Usually a difference in AIC higher than 2 demonstrates a unequivocal support for the model with the lowest AIC.”

We agree, and we described the AIC and the ΔAIC in the Material and Methods section. Also, we included a column of ΔAIC in Table 1 and we highlight the importance of the ΔAIC in the selection of models in the Discussion section. Please see the text highlight in blue in the manuscript.

2. “The discussion still contains sentences about AIC and LRT, which to me should be removed. Also, some sentences of the discussion repeats parts of the methods, which doesn't make sense. Discussion is still very lengthy to the amount of results you have and must be shortened. See my comments in the pdf attached. Avoid citing tables in the discussion.

We agree, and we removed the sentences about AIC and LRT in the discussion section. We removed the citations of Tables in the discussion section. Also, we re-written almost everything in the discussion section, trying to be clearer and objective. Please see the text highlight in blue in the manuscript.

3. (line 302-304) “and then what? What's the implication of this result? You're simply restating the result here in the discussion, when you should be indeed **explaining** them... Also, there's no citation in this paragraph...”

We re-written the discussion section. At this point we include a topic of variance components and genetic parameters. We discuss deeper about these parameters and their importance in maize breeding. Please see the text highlight in blue in the manuscript.

Thank you very much for the review and for the excellent contributions to improvement of this manuscript.

Reviewer #2:

“I have now reviewed the revised version of Coelho et al, in which the authors present a detailed multi-environmental trial analysis of 84 maize hybrids. As mentioned in my previous review, the goal of the paper is clear. The manuscript clearly aims to better understand the extent to which GEI interactions might be relevant to maize breeding programs. The approach is also sound. The authors use a simple model-fitting approach and directly compare the model fit of three main model classes (CSM, MTM and RRM).

Having said that, I have now read this paper multiple times and have postponed writing my review to gather my thoughts. As a reviewer, I strive to make suggestions that will allow authors to improve their manuscript. My sincere hope is that the revised version will be clearer and make a stronger argument in favor of the manuscript. In my view, the revised version of this manuscript does not succeed in addressing most of the comments raised by the editor and me in its previous iteration.”

1. “I still find the manuscript to have significant shortcomings in regard to organization/clarity and in placing the work in a greater context.”

We agree, and we made several changes/corrections in the manuscript. Please see the text highlight in blue in the manuscript.

2. “As I mentioned before, the goal of the manuscript is straightforward: compare these three model classes and infer the practical consequences of choosing one over the other (in terms of GEI). While the manuscript presents in detail the difference between these models in terms of their estimated components, it spends little to no time exploring the practical consequences of choosing one over the other.

2.1) “Yes, the parameters obtained by the different models are different, but what does that mean in practical terms?”

In practical terms, the best fitted model can provide more accurate estimates of genetic parameters and predicted genetic values, and consequently can maximize the efficacy of the breeding program. This is the first study that apply the RRM for analysis of MET in maize breeding and demonstrated the great advantage of this models in MET analysis.

2.2) “To what degree will it change the standard agricultural practice if the genotype rankings are so similar and if the metrics of selective accuracy are nearly identical between at least two of the models?”

The main advantage of RRM over the CSM and MTM is the ability to predict genotypic performance in environments where a genotype has not been evaluated (and this can increase the efficacy of breeding programs). Besides that, MTM tends to present problems in relation to convergence due to the large number of parameters estimated. Thus, the MTM became prohibitive when the number of environments is high.

2.3) “Also, as mentioned by the AE, what are the uncertainties around these parameters? Uncertainty in parameters is essential for a proper interpretation of the results. Should we observe a similar type of discrepancy between the three models if we were working with another population in different environments? In other words, the manuscript needs to go beyond reporting the technical details and expand on the biological implications.”

We added in Table 2 the standard errors of the variance component estimates. The results for another population evaluated in other environments can be different, but the statistical properties of the models will remain unchanged.

2.4) “As admitted in the main text of the manuscript, some of these models do not even have parameters that are capable of estimating GEI components directly (such as the MTM), despite that being the main goal of the manuscript.”

The MTM are the most complete models to deal with GEI. The fact that these models don’t be able to estimate the variance of the GEI interaction is not a limitation, because the GEI can be observed through the genotype ranking. Please see the text highlight in blue in the manuscript.

3) “Another issue that was not properly addressed was the use of figures.”

3.1) “Other than the addition of few red lines in one of the figures (Figure 1), there was no noticeable attempt to make the figures clearer. Why not highlight the crossing interactions (if that is the main argument being made) in Figure 1?”

Aiming to clarify the Fig 1, we split in A and B. In B, we reduced the number of reaction norms to make it easier to see the reaction norms and their intersections (what represent the presence of the GEI).

3.2) “Also, the figures came without caption this time. The figure with large matrices (which has no number) is very hard to read.”

We made some changes in the letter’s font, and we changed it to bold letter, to become it easier to read.

In short, the revised version of Coelho et al. remains a potentially interesting contribution to the field but falls short of making a stronger case at this point in time.

Thank you very much for the review and for the excellent contributions to improvement of this manuscript.

---

## [Editor Report · Decision Letter 2]

9 Nov 2020

Multiple-trait, random regression, and compound symmetry models for analyzing multi-environment trials in maize breeding

PONE-D-20-15130R2

Dear Dr. Bhering,

We’re pleased to inform you that your manuscript has been judged scientifically suitable for publication and will be formally accepted for publication once it meets all outstanding technical requirements.

Kind regards,

Diogo Borges Provete, PhD

Academic Editor

PLOS ONE
---

## [Editor Report · Acceptance letter]

11 Nov 2020

PONE-D-20-15130R2 

Multiple-trait, random regression, and compound symmetry models for analyzing multi-environment trials in maize breeding 

Dear Dr. Bhering:

I'm pleased to inform you that your manuscript has been deemed suitable for publication in PLOS ONE. Congratulations! Your manuscript is now with our production department. 

Kind regards, 

on behalf of

Dr. Diogo Borges Provete 

Academic Editor

PLOS ONE